# Geometric Constraints as General Interfaces for Robot Manipulation

## Abstract

We present GeoManip, a framework to enable generalist robots to leverage essential geometric constraints derived from object-part relations for robot manipulation. For example, cutting a carrot typically requires the knife's blade to be perpendicular to the carrot's medial axis. By capturing geometric constraints through symbolic language representations and translating them into low-level actions, GeoManip bridges the gap between natural language and robotic execution, boosting the generalizability across diverse, even unseen tasks, objects, and scenarios. Beyond vision-language-action models that require extensive training, GeoManip operates training-free by leveraging large foundational models: a constraint generator to predict stage-specific geometric constraints and a geometry parser to locate the involved object parts. A solver then optimizes trajectories for the inferred constraints from the task descriptions and scenes. Further, GeoManip learns in-context and provides five appealing human-robot interaction features: on-the-fly policy adaptation, learning from human demonstrations, learning from failure cases, long-horizon action planning, and efficient data collection for imitation learning. Extensive evaluations on both simulations and real-world scenarios demonstrate GeoManip's state-of-the-art performance, with superior out-of-distribution generalization while avoiding costly model training. Project website: https://sites.google.com/view/geomanip-anonymous

## 1 Introduction

Recent research Tang et al. (2024); Xu et al. (2024); Ko et al. (2023); Du et al. (2024); Bharadhwaj et al. (2024); Yuan et al. (2024); Baker et al. (2022); Chen et al. (2021); Huang et al. (2024b); Liang et al. (2023b); Huang et al. (2023b); Duan et al. (2024a) on utilizing vision-language models (VLMs) to develop general robot manipulation policies has drawn much attention, leveraging their vision understanding Tang et al. (2024); Xu et al. (2024); Ko et al. (2023); Du et al. (2024); Bharadhwaj et al. (2024); Yuan et al. (2024); Baker et al. (2022); Chen et al. (2021) and language reasoning Huang et al. (2024b); Liang et al. (2023b); Huang et al. (2023b); Duan et al. (2024a) abilities. Such language-based methods offer benefits like providing rich contexts for generalizable and interpretable policies, enabling step-by-step reasoning, and allowing on-the-fly modifications for robotic control. However, language is conceptual, lacking inherent information on the 3D geometry, Hence, it is hard to generate precise low-level robot actions.

Vision-language-action models (VLAs) Kim et al. (2024); Liu et al. (2024b) have emerged as recent solutions that implicitly bridge perception, reasoning, and execution in an end-to-end manner, equipping robots with the ability to plan low-level actions to follow human instructions with contextual awareness. However, they rely on large-scale and task-specific data for training and often struggle to generalize to novel tasks or objects. Besides, the language-to-action conversion is a "black box" without interpretability. To solve these challenges, a natural solution is to explicitly model the environment and 3D geometry by developing an intermediate representation that can be articulated in a high-level language while accurately depicting low-level actions, effectively connecting natural language and robotic actions.

To this end, we propose to introduce object-centric geometric constraints as an interface to connect language instructions with precise robot actions. Leveraging the reasoning capabilities in vision-language models (VLMs), these constraints can be defined by natural language and then converted

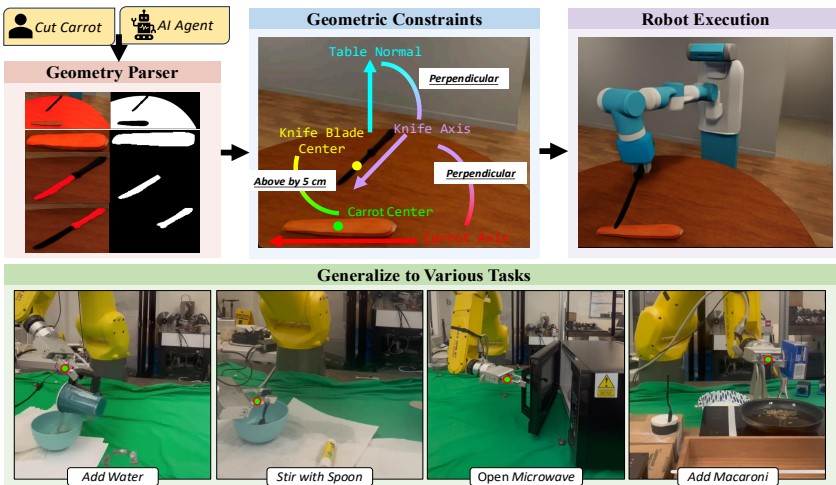

Figure 1: We propose to derive geometric constraints to bridge the gap between high-level language descriptions and low-level robot actions. Top: an example of the carrot cutting task. Down: as demonstrated experimentally, GeoManip is able to execute diverse tasks in general settings.

into symbolic forms to provide accurate spatial-aware guidance to producing low-level trajectories for the task. See Fig. 1 for an example, in the task "cut the carrot with the knife" to lift the knife above the carrot, the geometric constraints are: (i) the heading direction of the knife blade must be parallel to the table surface (perpendicular to the normal of the table surface); (ii) perpendicular to the carrot's axis; and (iii) the center of the knife should be positioned around 5 cm above the center of the carrot.

In this work, we present GeoManip, a framework for building generalist robots that can leverage geometric constraints as an interface to generate precise manipulation trajectories. Given a task described in natural language and the current scene observation, GeoManip decomposes the task into sub-tasks, ensuring satisfiable geometric constraints within each sub-task and consistent and smooth transitions between sub-tasks. Specifically, GeoManip comprises (i) a geometry parser that identifies object parts where geometric properties can be defined, via a proposed *select-process scheme*; (ii) a constraint generator that uses geometric knowledge to produce symbolic constraints and cost functions for each step; and (iii) a cost function-based trajectory solver that minimizes the overall costs, i.e., constraint violation, through optimization. Notably, code generation is integrated to process selected masks in the geometry parser and represent cost functions in the constraint generator.

With the careful design, GeoManip can naturally serve as a robot generalist capable of (i) on-the-fly policy adjustments based on human language feedback, (ii) learning from failure cases and adjusting the policy accordingly, (iii) learning from human demonstrations, (iv) performing long-horizon tasks via decomposition, and (v) efficient data collection for imitation learning. Experiments conducted in both virtual environments, such as MetaWorld Yu et al. (2020) and Omnigibson Li et al. (2023a), as well as real-world scenarios, demonstrate GeoManip's wide applicability, training-free, and out-of-distribution (OOD) generalizability across diverse object types, positions, and poses.

To summarize, our contributions are three-fold:

- We propose a novel generalist robotic framework GeoManip, using geometric constraints as an interface for robotic manipulation. It is simply driven by high-level language instructions with two key designs: a geometry parser with the select-process scheme and a constraint generator for cost-based planning.

- GeoManip is capable of reasoning about task constraints with five appealing benefits for robot learning, including learning from human demonstrations, on-the-fly policy adaptation, learning from failure cases, long-horizon planning, and efficient data collection for imitation learning.

- Extensive experiments in both simulations and the real world demonstrate GeoManip's effectiveness and generalizability, even to OOD scenarios with no training efforts needed.

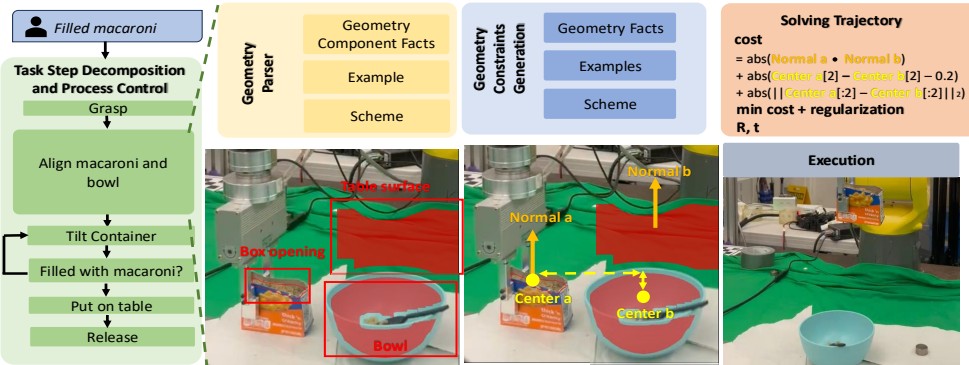

Figure 2: Given the user's task description, our method decomposes the task into multiple sub-tasks and forms the process control. For each stage, we first design a geometry parser to segment and obtain the point cloud for relative geometric components. Then, we develop a geometry constraint generation module to generate constraints among the geometric components that are necessary to complete the sub-task. Finally, we establish the cost functions to measure the fulfillment of the geometric constraints and solve the robotic trajectories via optimization.

## 2 RELATED WORK

**Robot Manipulation with Large Models.** There are three branches of work that manage to harness large models for robot manipulation. The first branch of work trains or fine-tunes vision-language-action model (VLA) with action-annotated data Brohan et al. (2022; 2023); Walke et al. (2023); Ebert et al. (2021); Huang et al. (2023a); Li et al. (2023b); Zhen et al. (2024); Driess et al. (2023); Chen et al. (2024); Kim et al. (2024), visual affordance data Li et al. (2024); Huang et al. (2024a); Yu et al. (2024) or motion tokens Chen et al. (2024) to achieve end-to-end action predictions given observations. The second branch of work Huang et al. (2023b); Duan et al. (2024b); Liu et al. (2024a); Huang et al. (2024b); Wang et al. (2024); Liang et al. (2023b); Ahn et al. (2022); Mu et al. (2024a); Song et al. (2023); Mu et al. (2024b); Zawalski et al. (2024) uses VLM as a high-level planner and divides each manipulation task into multiple sub-goals. The third branch of work Huang et al. (2023b; 2024b); Liang et al. (2023b) uses VLM to reason object relations or generates codes to derive low-level trajectory. However, most approaches, including the recent work ReKep Huang et al. (2024b), focus primarily on key-point relations. Our work lies in the third branch of work. Distinctively, we formulate geometric relationships for more precise robotic manipulation guidance.

**Spatial Relation Constraints for Robot Manipulation.** Spatial relations can be formulated as constraints for robot manipulation. Some recent works Kingston et al. (2018); Ratliff et al. (2009); Schulman et al. (2014); Sundaralingam et al. (2023); Marcucci et al. (2024); Ratliff et al. (2018) model the manipulation as an optimization problem and solve the constraints globally with various solvers to achieve the desired goal. Some other works Toussaint (2015); Toussaint & Lopes (2017); Toussaint et al. (2018); Xue et al. (2024) perform task decomposition and incorporate multi-stage spatial relationship. Recently, Huang et al. (2023b; 2024b) uses the VLM to reason object spatial relation automatically, given a task description. Some works Zhen et al. (2024); Zawalski et al. (2024) simply capture the spatial relationship among objects during the model finetuning or inference, thereby empowering the model with spatial awareness to improve its performance for action prediction.

**Open-vocabulary Object Detection and Part Segmentation.** Vision tasks in an open-vocabulary setting is challenging due to limited data. Despite their different model and training pipeline designs, most of the works Gu et al. (2021); Cheng et al. (2024); Du et al. (2022); Kuo et al. (2022); Wu et al. (2023); Zhong et al. (2022) address open-vocabulary object detection by aligning text embeddings and visual features of local regions. Open-vocabulary part segmentation is even more challenging for the countless part categories. Most methods either make use of the similarity of semantic and visual features Wei et al. (2024); Sun et al. (2023); Ding et al. (2023) or finetune the VLM model with part-level segmentation annotation data and harness its reasonability Lai et al. (2024); Zou et al. (2023); Wang & Ke (2024). However, they still cannot achieve satisfactory performance for OOD object-part segmentation.

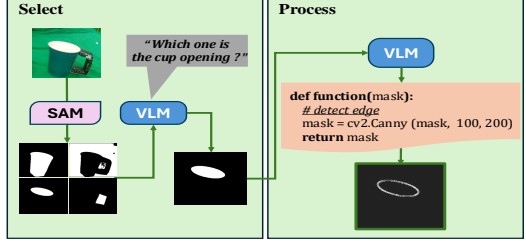

Figure 3: Existing methods (LISA Lai et al. (2024), OV-seg Liang et al. (2023a)) fail to segment the fine-grained geometric components, while our method succeeds.

Figure 4: Illustration of our select-process scheme. Our method first selects the most relevant mask for the target object part and then further processes and refines it.

## 3 METHODS

Given the current scene observation and a human language instruction, we propose the GeoManip framework, utilizing the geometric constraints among objects as an interface to generate manipulation trajectories to accomplish the task. Please refer to Fig. 2 for an illustration. The GeoManip framework consists of four steps. First, we decompose the task into multiple sub-tasks that are completed step-by-step and we create a process control for more complex tasks. Second, for each sub-task, we present a novel *select-process* solution to identify the relevant geometric components, which are fine-grained object-part structures that help define the spatial relationships among objects. Third, we generate the geometric constraints needed to accomplish the task based on the identified geometric components and some fundamental geometric principles that we provide. Last, we convert the geometric constraints into cost functions to guide trajectory planning in robotic manipulation through an optimization. Note that since the first and the third steps can be learned in an in-context fashion, they can be generated interactively, providing 5 important features.

In the following, we introduce the task decomposition and process control in Sec. 3.1, present the geometric parser in Sec. 5, and provide details on the constraint generation module in Sec. 3.3. In addition, we present the cost function and trajectory generation process in Sec. 3.4. Finally, we present the designs and examples to achieve the five generalist features separately in Sec. 5, after representing our experiment results in Sec. 4.

### 3.1 TASK DECOMPOSITION AND PROCESS CONTROL

Fig. 2 illustrates our process control. We leverage the VLM's capability for task composition to divide a task into multiple sub-tasks. For example, the task "Filled macaroni" can be decomposed into six sub-tasks. For many simple tasks, the sub-tasks are executed sequentially till the end of the task. However, complex tasks require loop or branch control. For example, to pour a certain amount of water from a cup into a container, we should repeatedly tilt the cup and check if the target container has enough water until the desired amount is reached. This motivates us to add process control in the task decomposition. To achieve this, we ask the VLM to decide the next sub-task to transit to upon finishing the current one. For example, to "check if the pan is filled with macaroni," we allow the VLM to capture previous RGB images to check if the conditions are met.

### 3.2 GEOMETRY PARSER

Using the language instruction and the current scene observation as input, we identify the fine-grained geometric components. A geometric component is a part of the object, on which a geometry can be clearly defined. For example, "cup opening" is a geometric component, where we can clearly define the plane across it, and "spoon tip" is another geometric component, where we can clearly define its center point. These geometric components are necessary for geometric constraint analysis.

However, all existing open-vocabulary image segmentation methods Lai et al. (2024); Liu et al. (2023); Wei et al. (2024) fail to identify geometric components. They may output the entire object or an incomplete object part, as illustrated in Fig. 3.

Observing that it is relatively easier to perform class-agnostic segmentation and that existing VLMs have an extraordinary ability to understand visual concepts, we combine these two to introduce a select-process scheme to tackle the geometry parsing task.

The select-process scheme consists of two steps, as shown in Fig. 4. First, we capture an image $I$ of the scene and leverage the Segment-Anything Model (SAM) Kirillov et al. (2023) to obtain the class-agnostic masks, i.e., $\{M\}_1^N = SAM(I)$. We further query the VLM to select the most accurate mask. Specifically, we provide a language description of the geometric component and pair the image $I$ with each mask to form $\{(I, M)\}_1^N$ to aid the selection. However, even after selecting the best-matched mask, it may not accurately represent the geometric component. Hence, we further leverage the VLM to refine the selected mask to represent the geometric component. Let $M^*$ be the selected mask, we use $M^*$, its corresponding image $I^*$, and the geometric part description to query the VLM to implement a code function $g : \mathbb{R}^{H \times W} \to \mathbb{R}^{H \times W}$ to generate processed mask $M' = g(M^*)$. Please refer to Fig. 4 for an illustration of the processing procedure. The detailed prompt for querying VLM to select and implement code can be viewed in Appx. H.5 and Appx. H.6.

We observe that our method can generate more accurate segmentation results to represent the geometric components as illustrated in Fig. 3.

## 3.3 CONSTRAINT GENERATOR

The constraint generation module infers the geometric constraints among geometric components that are required to complete the current sub-task. The geometric constraints are defined on the geometric components and describe the spatial relationships among components, e.g., parallel, perpendicular, directly above, to the left by 10 cm, etc.

Given the set of geometric components "{GeoComp 1, GeoComp 2, . . . }" involved in the sub-task and a language description of the constraint "ConsDesc", a geometric constraint can be formulated as a tuple ( {GeoComp 1, GeoComp2, . . . }, ConsDesc). For example, for the stage to align a knife with a carrot ready to be cut, the geometric constraints are:

- ({"the knife blade", "the carrot"}, "the heading of the knife blade is perpendicular to the axis of carrot"")
- ({"the knife blade", "the table surface"}, "the plane of the knife blade is perpendicular to the plane of the table surface"")
- ({"the knife","the carrot"}, "the center of the knife is directly above 'the center of the carrot by 10 cm"")

Following Huang et al. (2024b), we further specify if the constraint is a sub-goal constraint (needs to be satisfied only at the end of the trajectory), or a path constraint (needs to be satisfied throughout the entire trajectory).

We harness the strong language reasoning ability of VLM to generate geometric constraints automatically. To achieve this, we need three types of components in the prompt. The first is the geometry principles which include some basic geometry facts such as "To be perpendicular to a plane is to be parallel to its normal". The second is the output rules indicating how the geometric constraint should be formulated. Finally, we include some concrete examples for the VLM to follow. Details of the prompts for constraint generation are shown in Appx. H.1 and Appx. H.2.

## 3.4 COST FUNCTIONS AND TRAJECTORY GENERATION

We develop cost functions to quantify the satisfaction of geometric constraints during robotic manipulation, which are used to guide the gripper pose to complete the sub-task. Therefore, we propose to use the VLM to generate codes that represent the cost functions based on the language format of the geometric constraints, leveraging the code generation ability of the VLMs.

More specifically, we ask the VLM to generate a code function $f : \mathcal{P} \to \mathbb{R}^+$ for each cost constraint. The function $f$ takes the set of geometric component's point clouds $\mathcal{P} = \{\mathbf{p}_1, \mathbf{p}_2, \cdots\}(\mathbf{p}_i \in \mathbb{R}^{N_i \times 3}$ which is the point cloud of geometric components $i$) as input and outputs a non-negative floating value representing the degree of violation with the geometric constraint (lower is better), and the minimum value of 0 is reached when the geometric constraint is perfectly satisfied.

For the VLM to generate the function correctly, we need to provide three components in the prompt as follows: 1. The rules and format for the output. 2. Examples of (geometric constraint, and cost function). 3. General basic geometric facts such as how to orbit or rotate points around an axis. The details of the prompt can be viewed in Appx. H.3 and Appx. H.4. We obtain a cost function for every geometric constraint, forming a set of path cost functions $\mathcal{F}^p$ for the path constraints and a set of sub-goal path functions $\mathcal{F}^s$ for the sub-goal constraints.

We leverage these cost functions to guide robotic motions by generating the manipulation trajectories to satisfy the geometric components. First, we identify which geometric component is manipulated by the robotic gripper by finding the ones belonging to the grasping object, we denote the set of point clouds of moving components as $\mathcal{P}^m$. We further denote the set of stationary geometric components' point clouds as $\mathcal{P}^s$. Since the gripper is rigidly attached to the moving component $\mathcal{P}^m$, they share the same transformation. Hence, solving the gripper's target 3D rotation matrix $\mathbf{R} \in SE(3)$ and transformation vector $\mathbf{t} \in \mathbb{R}^3$ for the sub-goal constraints is equivalent to solving the following optimization problem:

$$\min_{\mathbf{R},\mathbf{t}} \frac{1}{K^s} \sum_{f \in \mathcal{F}^s} f(\mathcal{P}^s \cup (\mathbf{R}\mathbf{R}_0^{-1} \bigotimes (\mathcal{P}^m \bigoplus -\mathbf{t}_0) \bigoplus \mathbf{t})) \tag{1}$$
$$+ \alpha \|\mathbf{t} - \mathbf{t}_0\|_2 + \beta \|euler(\mathbf{R}\mathbf{R}_0^{-1})\|_1,$$

where $\mathbf{R}_0$ and $\mathbf{t}_0$ are the gripper previous rotation matrix and translation vector respectively, while $\mathbf{R}$ and $\mathbf{t}$ denote the optimized rotation matrix and translation vector. $euler(\cdot)$ is the operation to get the Euler angle in three rotation axes from the matrix. $\bigoplus$ and $\bigotimes$ are vector addition and matrix product for each element in the set $\mathcal{P}^m$. $\alpha$ and $\beta$ are two scalars to regularize in translation and rotation, respectively. After the target rotation $\mathbf{R}$ and the translation $\mathbf{t}$ of the gripper are obtained, we further extract the entire manipulation trajectory. We first interpolating between ($\mathbf{R}_0$ and translation $\mathbf{t}_0$) and ($\mathbf{R}$ and translation $\mathbf{t}$) and generating several "control points" between. For each "control points", we optimize it using $\mathcal{F}^p$ in a similar way as Eq.1.

## 4 EXPERIMENTS

### 4.1 IMPLEMENTATION DETAILS

**Technical Details for the VLM design.** We use GPT-4o OpenAI (2024) as the VLM for our implementation. We use the SLSQP algorithm Kraft (1988) for optimization solving. For optimization, we set $\alpha = 0.02$, and $\beta = 0.075$ for regularization. We use Grounding-DINO Liu et al. (2023) to locate and crop the target object first before processing it with the geometry parser to prevent overwhelming mask candidates generated by SAM Kirillov et al. (2023).

**Virtual Benchmarks.** We perform experiments on two virtual environments: Meta-World Yu et al. (2020) and OmniGibson Li et al. (2023a), including 10 diverse tasks. The Meta-World environment is a simulated benchmark featuring numerous predefined tasks for reinforcement learning. We compare the performance of our method on six tasks, following the common settings Tang et al. (2024); Ko et al. (2023). The OmniGibson environment is another virtual environment. It features realistic physics simulation and rendering and enables user-defined tasks. We further develop 4 tasks on OmniGibson to test our method.

**Real-world Environment.** In addition to the experiments on the simulators, we design 4 more tasks to demonstrate the effectiveness of our method in real-world robotic settings. A table is set up with objects placed on top of it. A RealSense D435i camera is set up for visual sensing and a FANUC LR mate 200id robot arm is equipped to perform the task.

**Evaluation Metrics.** For all benchmarks, we assess the performance of all methods based on each task's success rate (number of success trials/number of total trials).

### 4.2 RESULTS ON VIRTUAL BENCHMARKS

**Meta-World Environemnt.** We first evaluate the performance of the methods on six tasks in the Meta-World benchmark Yu et al. (2020). We follow the common settings Tang et al. (2024); Ko et al. (2023) and only consider gripper translation in this environment. For each task and each camera view, we evaluate our method 5 times, and at the start of each trial, we randomize the initial poses and positions of the object involved in the task. For objects that are too small to be identified by the VLM model, we use the ground-truth mask. We test each task for 3 camera positions and report the best result across different views. We report the success rates and provide an overall success rate across all tasks.

We compare our method with seven state-of-the-art methods, i.e., BC-Scratch Nair et al. (2022), BC-R3M Nair et al. (2022), UniPi Du et al. (2024), Diffusion Policy Chi et al. (2023), AVDC Ko et al.

Table 1: **Visual Illustration of each Task and Results on the Meta-World Dataset.**

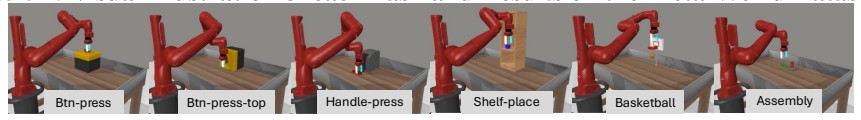

|  | basketball | shelf-place | btn-press | btn-press-top | handle-press | assembly | **overall** |
|---|---|---|---|---|---|---|---|
| BC-Scratch | 21.3% | 36.0% | 0.0% | 0.0% | 34.7% | 0.0% | 15.3% |
| BC-R3M | 0.0% | 0.0% | 36.0% | 4.0% | 18.7% | 0.0% | 9.8% |
| UniPi | 0.0% | 0.0% | 6.7% | 0.0% | 4.0% | 0.0% | 1.8% |
| Diffusion Policy | 8.0% | 0.0% | 40.0% | 18.7% | 21.3% | 1.3% | 14.8% |
| AVDC | 37.3% | 18.7% | 60.0% | 24.0% | 81.3% | 6.7% | 38.0% |
| SceneFlow | **96.0%** | 29.3% | 50.7% | **96.0%** | 40.0% | **46.7%** | 59.8% |
| Pi0 | 0.0% | 0.0% | 20.0% | 6.7% | 20.0% | 0.0% | 7.8% |
| **Ours** | 73.3% | **60.0%** | **80.0%** | 73.3% | **100.0%** | 40.0% | **71.1%** |

Table 2: **Visual Illustration of each Task and Results on the Omnigibson.**

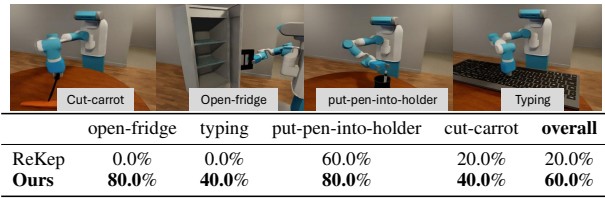

|  | open-fridge | typing | put-pen-into-holder | cut-carrot | **overall** |
|---|---|---|---|---|---|
| ReKep | 0.0% | 0.0% | 60.0% | 20.0% | 20.0% |
| **Ours** | **80.0%** | **40.0%** | **80.0%** | **40.0%** | **60.0%** |

(2023), SceneFlow Tang et al. (2024), and Pi0 Black et al. (2024). Note that all compared methods require an additional training stage to learn the robotic action following the training scheme referred to Ko et al. (2023), while our method is training-free.

We present visual illustrations of each task along with the results of our method and the comparison methods in Table 1. From the table, we can see that our method greatly outperforms BC-Scratch, BC-R3M, UniPi, AVDC, and SceneFlow by over 11% in terms of average accuracy. The results demonstrate the effectiveness of our method. Detailed input instructions visualizations for each task's complete execution can be found in Appendix A.

**OmniGibson Environment.** We evaluate our method on four tasks which require both translation and rotation of the gripper, further showing the effectiveness of our method. The detailed instructions of each task can be viewed in Appx. B.

We conduct 5 trials for each task and report the success rates to evaluate the methods. We also compare our method with a very recent work evaluated on the OmniGibson environment, the ReKep Huang et al. (2024b). The ReKep proposes to plan robotic manipulation based on the spatial relations among the key points on objects.

The visual illustration of each task along with the experimental results of our method and the compared method are shown in Tab. 2. From the table, we can see that our method consistently outperforms ReKep in each task. It outperforms ReKep by at least 20% in each task and achieves a 40% higher overall success rate. This is because our method better models the relations between objects via geometric constraints, which is more detailed and precise compared with the object's key points.

### 4.3 EXPERIMENTS ON REAL ENVIRONMENT

Furthermore, we test our method in real-world settings using four tasks: (i) picking and placing one object onto another object, (ii) pouring something from one container to another, (iii) opening/-closing an object with rotation / prismatic movement, and (iv) stirring something in a container. In Appendix C, we outline the criteria for task success.

For each task, we report the success rate over 10 testing sequences with randomized object types and initial poses. We compare our method to the baseline OpenVLA Kim et al. (2024), a training-dependent behavior-cloning approach, using 30 training trials for its training. We visualize some of our demonstration sequences and provide statistical results in Tab. 3. From the results, we can see that since our method can understand the geometric constraint, it can successfully manipulate various tasks involving different object types. By using the geometric constraint as the abstract interface, our method achieves at least a 40% higher success rate than OpenVLA, leading to an overall success rate that is 50% greater, while it is training free.

Table 3: **Visualization and Statistical Results on the Real-World Setting.**

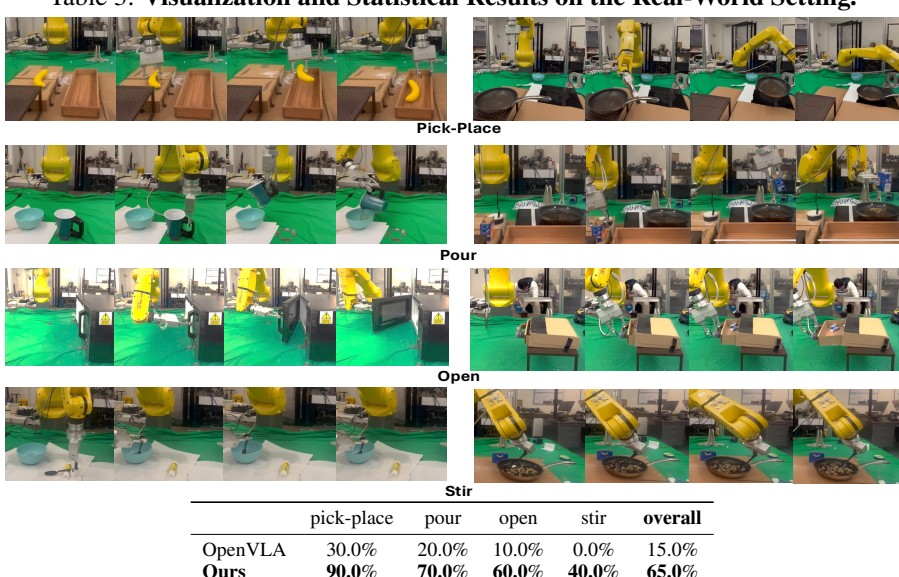

Pick-Place

Pour

Open

Stir

|  | pick-place | pour | open | stir | **overall** |
|---|---|---|---|---|---|
| OpenVLA | 30.0% | 20.0% | 10.0% | 0.0% | 15.0% |
| **Ours** | **90.0%** | **70.0%** | **60.0%** | **40.0%** | **65.0%** |

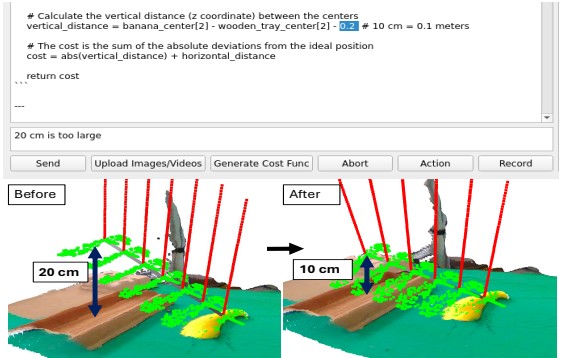

Figure 5: Example of our agent adjusting geometric constraints via human feedback. Red lines show the gripper's approach direction; green indicates the binormal. Initially, the agent plans to lift the banana 20 cm above the box (left), and it reduces the height to 10 cm after "the height is too large" feedback (right).

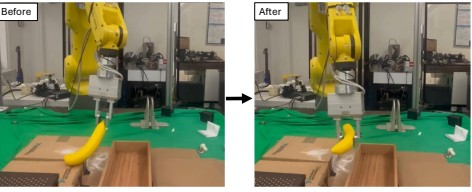

Figure 6: Example of our agent in learning from the failure case. Using the original constraints (blue in the black box below), the robot grasps the banana in an unsafe pose biased from the banana's center and the task fails with banana slipping from the gripper (bottom-left image). After prompting the embodied agent with the failure video, a new constraint is added (red in the black box below), enabling the robot to successfully grasp the banana at its center (bottom-right image).

## 5 GENERALIST EMBODIED AGENT FOR ROBOTIC MANIPULATION

Since our method uses geometric constraints as the interface to bridge the high-level planning and the low-level action, it can further be used to develop a generalized embodied agent for robotic manipulation. The detailed design of the AI interface and its usage is included in the Appx. D. Our embodied agent facilitates open-ended conversations with the user while generating geometric constraints, enabling users to provide further instructions. Video inputs are also allowed so that the agent can learn geometric constraints or observe failures from them. These designs empower GeoManip with five features for robotic manipulation: On-the-fly policy adaptation, learning from failure cases, long-horizon manipulation, learning from human demonstration, and data collection for training models.

### 5.1 ON-THE-FLY POLICY ADAPTATION

Since our embodied agent allows the user's open-ended conversation in generating the geometric constraints, the user can augment with further demands to adjust the geometric constraints. Specifically, the user can specify the distances or rotation angles when manipulating the object or input

```
Stage1:  Grasp the banana
- Original constraints
  - <"grasp", "the body of the banana">
  - <"sub-goal constraints":  "heading direction of the gripper approach", "plane of table surface", "heading direction of
  the gripper approach parallel to normal of plane of table surface">
  - <"sub-goal constraints":  "heading direction of the gripper binormal", "banana", "heading direction of the gripper binormal
  perpendicular to banana axis">

- New Constraints:
- <"sub-goal constraints", "gripper center", "banana center", "gripper center is aligned with banana center">
```

high-level commands. Then, the embodied agent responds to the user's input and adjusts the geo-
metric constraint. For example, by specifying the height above the wooden tray, the robot lifts the
banana above the wooden tray with a height modified from 20 cm to 10 cm as illustrated in Fig. 5.

## 5.2 LEARN FROM FAILURE CASES

Our method can learn from previous failure executions and improve the current manipulation policy.
To achieve this, the user uploads recordings of the robot's failed execution to the embodied agent,
accompanied by language commands like "Why did the robot fail to execute?" and "How can we
adjust the geometric constraints to improve performance?". The embodied agent then refines the
geometric constraints. We showcase an example in Fig. 6. The robot fails to place the banana into
the wooden tray because of the unsafe grasp position. By asking the embodied agent "The robot fails
and the banana slips", it refines the geometric constraint and adds an extra sub-goal constraint during
the grasp stage to enforce the grasp position to be close to the center of the banana. After re-solving
for the trajectory, the banana is safely grasped and transferred to the wooden tray successfully.

## 5.3 LONG-HORIZON TASKS

Our embodied agent can also handle long-term tasks by first asking the agent to divide long-term
tasks into multiple single tasks. We provide an example of a long-sequence demo for the task "Add
the pan with macaroni and water. Add salt with the spoon and stir the pan." in Fig. 11. Please refer
to the Appx. F for the complete video.

## 5.4 LEARN FROM HUMAN DEMONSTRATIONS

Our embodied agent learns from human demonstrations by summarising the geometric constraints
from the videos of human manipulation. As demonstrated in Fig. 12 in Appx. G, in the task of "open
the box", the original policy treats the box as the drawer and tries to move the flap of the box away
from the box's center. After viewing a human demonstration video, the agent correctly identifies the
box as open by lifting the lid. It then refines the sub-goal to lift and rotate the flap around the box
edge, successfully opening it.

## 5.5 DATA COLLECTION FOR REWARD MODEL

Since geometric constraints are consistent for the same manipulation task, we only need to generate
them once, regardless of object positions and orientations. Also, segmentation is performed once for
an object configuration, after which we use tracking models like CoTracker Karaev et al. (2023) for
other poses. This allows GeoManip to efficiently generate trajectories for varied initial configura-
tions, which can be used to train or fine-tune the VLA model or the reward model. Detailed settings
and experiments are provided in Appendix E.1.

## 6 LIMITATIONS

Our methods fails mainly when it fails to generate geometric components or the geometric con-
straint. For the former, we fail when 1. The geometric component is not clearly shown in the
camera. 2. The geometric component is not language-describable. For the latter, we fail when: 1.
The VLM model misses the critical geometric constraints. 2. The action is not language-describable.
We leave these issues to tackle in our future work.

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

# A    METAWORLD ENVIRONMENT

## A.1    TASK DESCRIPTIONS

We design 6 tasks for the MetaWorld environment:

- Btn-press. Task description: press the red button from its side. Success condition: The red button is entirely pressed.

- Btn-press-top. Task description: Press the red button from top-down. Success condition: The red button is pressed entirely.

- Handle-press. Task description: Press the red handle. Success condition: The handle is entirely pressed.

- Shelf-place. Task description: Put the blue cube onto the middle stack of the shelf. Grasp the blue cube and lift it vertically before moving to the middle stack of the shelf. Success condition: The blue cube is on the middle stack of the shelf.

- Basketball. Task description: Put the basketball onto the hoop. Lift the ball vertically and move over above the hoop, Success condition: The basketball pass through the hoop.

- Assembly. Task description: Put the round ring into the red stick. Grasp the green handle of the round ring and put the hole into the red stick. Success condition: The red stick is inside the round ring.

## A.2    MANIPULATION VISUALIZATION

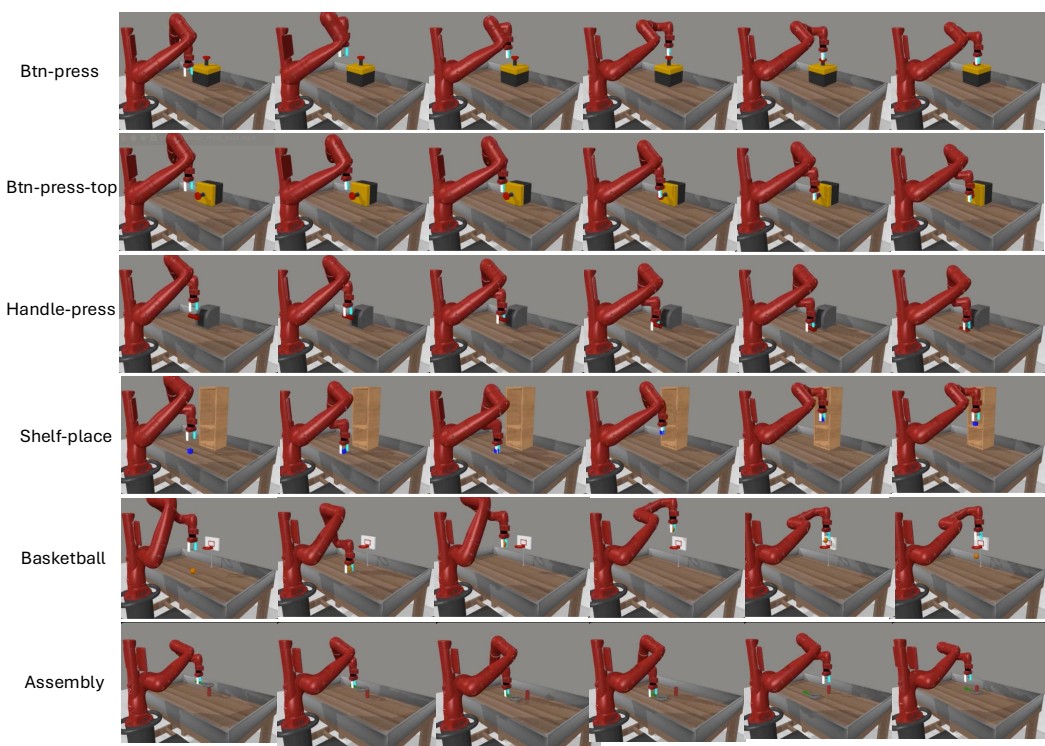

Figure 7: Visualization of execution for each task in MetaWorld environment.

# B    OMNIGIBSON ENVIRONMENT

## B.1    TASK DESCRIPTIONS

For Omnigibson environment, we design 4 tasks:

- Cut-carrot. Task description: Cut the carrot with the knife. Success condition: The knife blade intersect with the carrot top-down with its normal perpendicular to the carrot's heading direction.
- Open-fridge. Task description: Open the fridge. Success condition: The fridge door is open by at least 45 degrees.
- Put-pen-into-holder. Task description: Put the pen perpendicularly into the black cup. Success condition: The pen is inside the pen holder.
- Typing. Task description: Type "hi" on the computer keyboard. Success condition: The "H" key and the "I" key on the computer keyboard are pressed sequentially.

## B.2 MANIPULATION VISUALIZATION

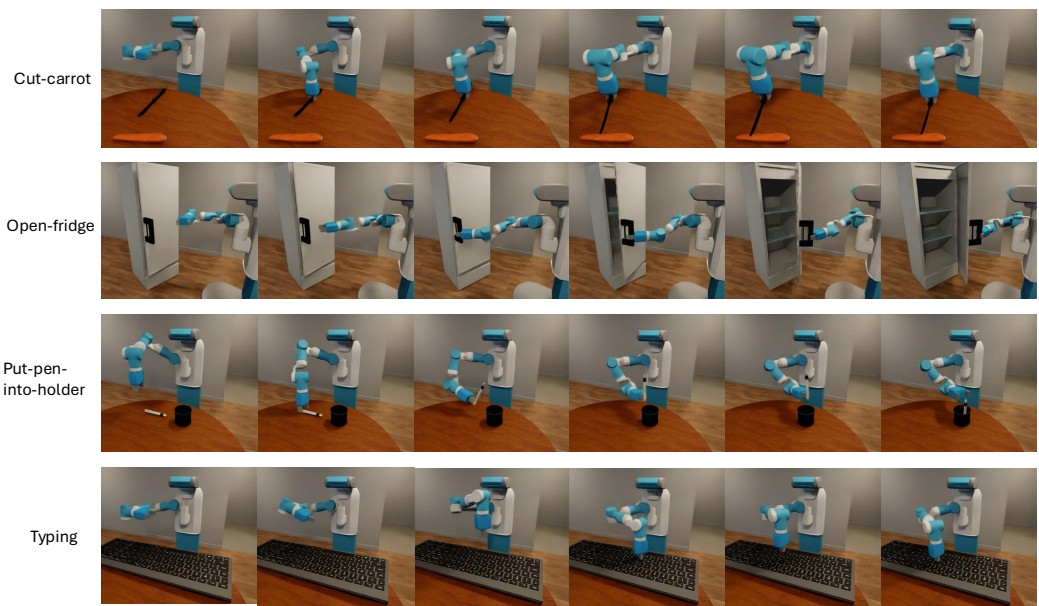

Figure 8: Visualization of execution for each task in Omnigibson environment.

## C REAL WORLD ENVIRONMENT

### C.1 TASK DESCRIPTIONS

We design 4 tasks for the real-world environment:

- Pick-place. Task description: Put <object A> into / onto <object B>. Success condition: <object A> is inside / on <object B>.
- Pour: Task description: Fill <object A> with <object B>. Success condition: <object B> is filled with some <object A>
- Open: Task description: open <object>. Success condition: <object> is open by at least 30 degrees / 5 cm.
- Stir. Task description: Stir <object A> with <object B>. Success condition: <object B> moves periodically inside <object A>.

## D DESIGN OF AI AGENT

See Fig. 9 for an illustration of our embodied agent interface, which consists of a user input block, a geometric constraint block, a cost function block, a geometric component visualizer, and a trajectory

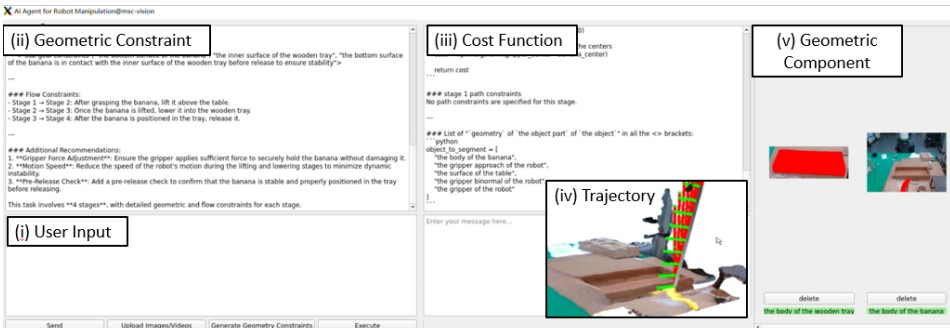

Figure 9: Our embodied agent comprises five components: (i) a user input block that accepts the current observation of the scene, the language command from user and uploaded videos of robotic to human manipulation of the sub-task; (ii) a geometric constraint block to display the generated geometric constraints for the sub-task allowing for modifications; (iii) a cost function block to present the developed cost function based on the geometric constraints; (iv) a geometric component visualizer to show the mask of the geometric component involved in the sub-task; (v) a trajectory visualizer that illustrates the planned trajectory in the scene.

visualizer. To accomplish a sub-task, the user uploads an image of the current observation of the scene, together with a language command. The generated geometric constraint generator generates geometric constraints and they are displayed in the geometric constraint block. The geometry parser identifies geometric components and they are displayed in the geometric component visualizer. The cost function generator generates cost functions according to the geometric constraints and they are displayed in the cost function block. The planned trajectory is generated by trajectory generator and it is displayed in the trajectory visualizer. What's more, the geometric constraints and cost functions can be generated interactively by providing descriptions, images, and videos in the user input block.

# E    DATA COLLECTION FOR TRAINING MODELS

In the following, we showcase how our method generates data for training robotic policies. Specifically, we show the performance of our training data collection scheme for the Vision-Language-Action (VLA) models and the reward models.

## E.1    DATA COLLECTION FOR VLA MODEL.

For the experiment, we apply this strategy to collect data for fine-tuning the OpenVLA model Kim et al. (2024) under two tasks: 1. Pick-stick: the robot needs to pick up the wooden stick and place it in the pan. The pan is large enough so that the task can be successfully achieved as long as the stick is above the pan. 2: Pick-banana: the robot needs to pick the banana and place it into a slim wooden box, which can only succeed if the banana axis is aligned with the long side of the wooden box. We collected 30 training data using our strategy, and we compared the performance of OpenVLA fine-tuning on our data and fine-tuning on manually collected ones. Note that our primary focus is on learning the manipulation trajectories, so we consistently use the ground-truth grasp pose in both settings to minimize interference from object grasping. We can see that the trajectories efficiently collected by our method (Ours) match a similar quality with the ones manually collected (Manual).

Table 4: Performance comparisons between Open-VLA trained with manually collected and data collected with our methods.

|  | Place-stick | Place-banana |
|---|---|---|
| Manual | 3/5 | 5/5 |
| **Ours** | 3/5 | 4/5 |

E.2   DATA COLLECTION FOR REWARD MODEL.

Furthermore, the generated cost function can indicate how close the current robotic state is to the target state for the manipulation task, the cost function itself can be viewed as a reward function. As a result, we can readily derive the reward value from the cost function to train a reward model that takes the current RGB observation $o$ as input. The model uses a simple ViT model as an encoder to encode the RGB image and uses a Multi-Layer Perceptron (MLP) to generate the reward score. During inference, we define a set of candidate actions: {Left, Right, Front, Back, Up, Down}, and each action moves the gripper positions along the corresponding direction by a small step. By denoting $o' = step(o, a)$ as the RGB observation after applying action $a$ on the previous scene with RGB observation $o$, we select the action $a$ according to $a = \operatorname*{argmax}_{a} R(step(o, a))$. We designed a naive example in which the robot needs to pick up the wooden stick. This concept and process is illustrated in Fig. 10.

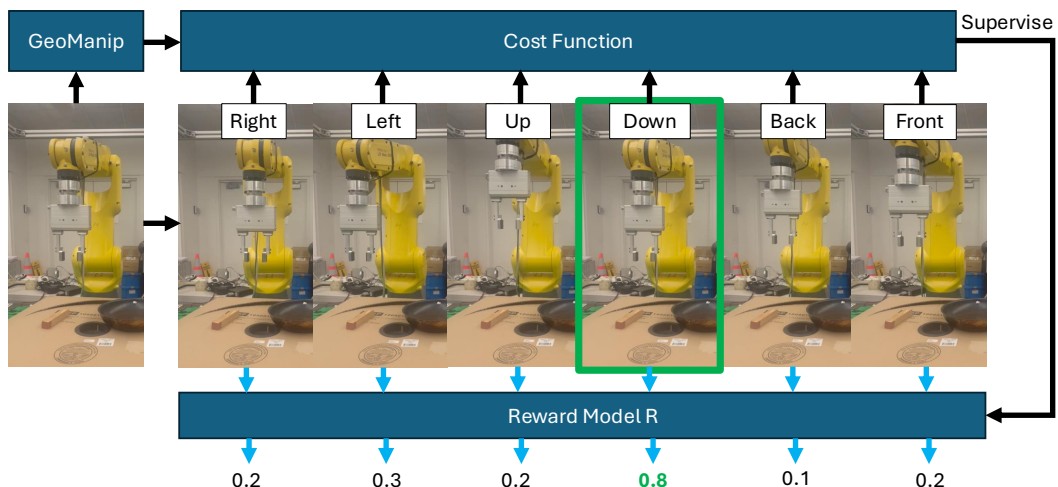

Figure 10: Example of using GeoManip to train the reward model $R$. The blue arrow is the data flow during inference. We select and execute the action that maximizes the reward (in this case, $a =$"Down").

## F   LONG-HORIZON TASK VISUALIZATION

We demonstrate two long-horizon tasks:

- Instruction: "Add the pan with macaroni and water. Add salt with the spoon and stir the pan." Execution demonstration video link: link.

- Instruction: "Add water to the plate, and heat the plate with the microwave." Execution demonstration video link: link.

The first task sequence is also visualized in Fig.11

## G   LEARN FROM HUMAN DEMONSTRATION

Our embodied agent learns from human demonstrations by summarising the geometric constraints from the videos of human manipulation. As demonstrated in Fig. 12, in the task of "open the box", the original sub-goal constraint treats the box as the drawer and the policy tries to move the flap of the box away from the box center. After including a human demonstration, it correctly captures the box as open by lifting the lid. Therefore, it refines the sub-goal constraint to lift and rotate the flap around the box edge, resulting in successful box opening.

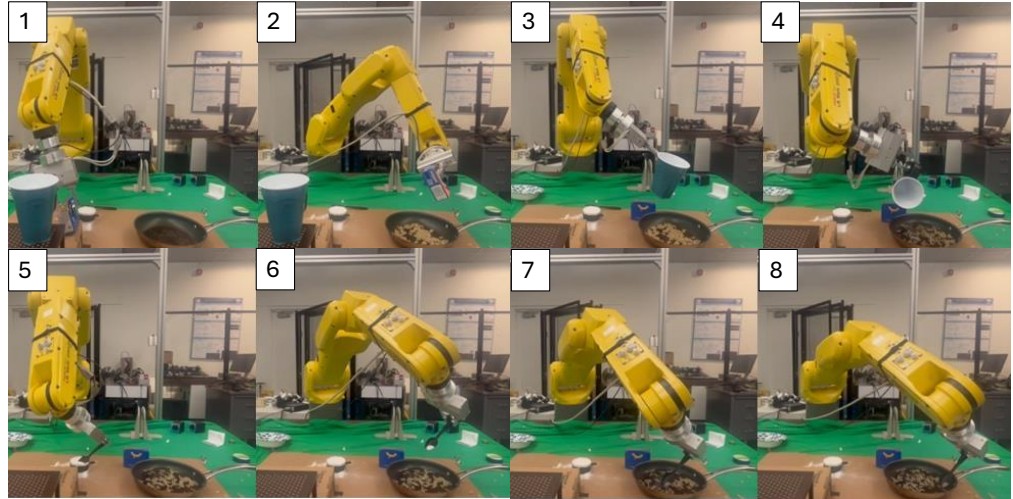

Figure 11: The embodied agent performing a long-sequence task.

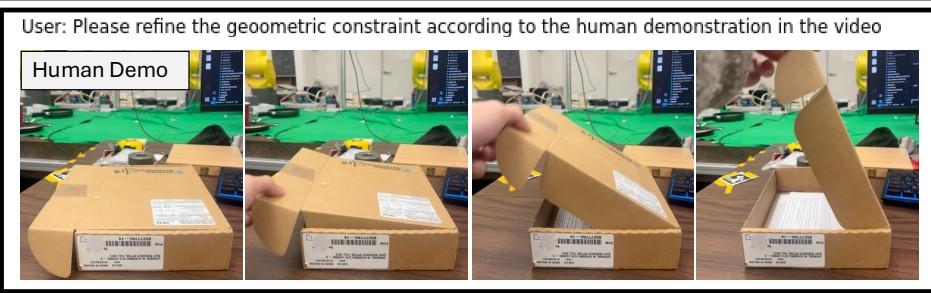

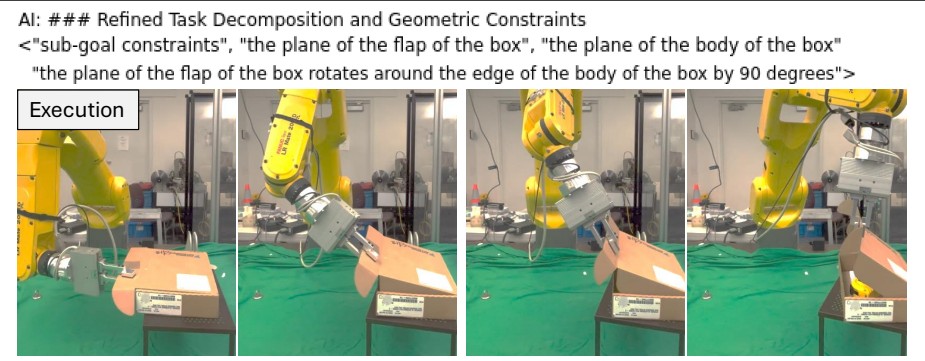

Figure 12: Example of the embodied agent learning from human demonstration for the task "open the box". Top: the original geometric constraint. Middle: the user requirement for learning from the video as well as the human demonstration video. Bottom: the refined geometric constraint after learning from a human demonstration, and the image sequence illustrates the execution process.

## H    PROMPTS

We design a scheme prompt and an example/knowledge prompt for each of the following modules: 1. task decomposition and process control. 2. geometry parser. 3. constraint generator and 4. cost function generation. The scheme prompt provides rules that the VLM model should follow to generate valid output that can be parsed. The example/knowledge prompts provide necessary

examples to follow or knowledge for in-context learning. For convenience in our implementation, we combine the prompts for module 1 and 2 into one, resulting in 6 prompts:

- Scheme prompt for task compositions and process control, and constraint generator.
- Example/knowledge prompt for task decompositions and process control, and constraint generator.
- Scheme prompt for cost function generation.
- Example/knowledge prompt for cost function generation.
- Scheme prompt for the geometry parser.
- Example/knowledge prompt for the geometry parser.

## H.1 SCHEME PROMPT FOR TASK DECOMPOSITIONS AND PROCESS CONTROL, AND CONSTRAINT GENERATOR

```
## Query
Query Task: "{Task Description}"

## Instructions
Suppose you are controlling a robot to perform manipulation tasks. The manipulation task is given as an
    ↪ image of the environment. For each given task, please perform the following steps:
1. Task decomposition and flow control:
Determine how many stages are involved in the task.
Grasping or releasing must be an independent stage.
Flow control controls the transition between stages
Some examples:
  - "pouring tea from teapot":
    - 6 stages: "grasp teapot", "align teapot with cup opening", "tilt teapot", "flow control: repeat '
        ↪ tilting teapot' until the cup if filled", "place the teapot on the table", "release"
  - "put red block on top of blue block":
    - 3 stages: "grasp red block", "drop the red block on top of blue block"
  - "reorientate bouquet and drop it upright into vase":
    - 3 stages: "grasp bouquet", "reorient bouquet", and "keep upright and drop into vase"
2. Geometric constraint and flow constraint generation: For each stage except for the grasping and release
    ↪ stage, please write geometric constraints and flow constraints in lines. Each line represents a
    ↪ constraint that should be satisfied.
 - Geometric constraint is a tuple of multiple elements: <"constraints type", "'geometry 1' of 'the object
    ↪ part' of 'the object', "'geometry 2' of 'the object part' of 'the object', ...(if any), "
    ↪ constraints">, each element is explained in the follows:
  - "geometry": Basic geometric primitive like the left edge, the center point, the plane, the normal, the
    ↪ right area, heading direction, and etc..
  - "the object part": the key object part on an object, like the tip, the opening, the handle, the hinge,
    ↪ the slider, the gripper, etc.
  - "the object": the complete object, like the black cup, the second door, the teapot, the robot, etc.
  - "constraint":
    - 1. basic geometric relationship including parallel, perpendicular, vertical, intersect, and etc..
    - 2. positional constraint like above, below, to the left / right, and etc..
    - 3. Distance range like "by 10 centimeters", "around 10 centimeters", "more than 25 centimeters", "10
        ↪ centimeters to 20 centimeters", "45 degrss", etc..
    - 4. Transformation like "rotate", "shift", etc.
  - Specify the <'geometry' of 'the object part' of 'the object'> in the "constraint"
  - "constraints type":
    1. "sub-goal constraints": constraints among 'geometry 1', 'geometry 2', ... that must be satisfied **at
        ↪ the end of the stage**. In other word, it specifies the constraints of the destination
        ↪ position.
    2. "path constraints": constraints among 'geometry 1', 'geometry 2', ... that must remain satisfied **
        ↪ within the stage**. In other word, it specifies the constaints on the way to the destination
        ↪ position.
 - Flow constraint is a tuple of multiple element <"flow constraint", "condition"> (goto stage ? if satisfied
    ↪ ; goto stage ? if not satisfied)
 - For each stage, there can be ONLY one flow constraint. If there are multiple flow constraint, use
    ↪ standalone stage to place these flow constraints
 - Do not ignore "of". There must of at least two "of": "'geometry' of 'the object part' of 'the object'". If
    ↪ you what to specify 'geometry' of the whole object, use 'geometry' of the body of 'the object'
 - For the grasping stage, sub-goal constraint 1 should be <"grasp", "the area of 'the object part' of 'the
    ↪ object'">
 - For grasping stage, you can also specify the sub-goal constraints of the heading direction of the gripper
    ↪ approach of the robot or the heading direction of the gripper binormal of the robot:
  - approach: the direction of the robot gripper pointing at, usually perpendicular to some surface. You can
    ↪ get the gripper approach by calling get_point_cloud("the gripper approach of the robot", -1). To
    ↪ find its heading direction, find its eigenvector with max eigenvalue.
  - binormal: the direction of gripper opening / closing, usually perpendicular to some axis / heading
    ↪ direction or parallel to some normal. You can get the gripper binormal by calling get_point_cloud
    ↪ ("the gripper binormal of the robot", -1). To find its heading direction, find its eigenvector
    ↪ with max eigenvalue.
 - To close the gripper only without grasping anything, output <"grasp", "">
 - If you want to use the gripper, only specify its center position, the heading direction(approach), or the
    ↪ binormal.
 - For the releasing stage, sub-goal constraint should be <"release">
 - Avoid using the part that is invisible in the image like "bottom", "back part" and etc.
 - Please give as detailed constraint as possible.
 - To move something, you must grasp it first.
 - Each stage can only do a single action one time.
```

```
1026
1027    – Don't omit stages for the repeating stages, expand and list them one by one.
        – Please answer according to the image we provided, which is the previous scene.
1028
1029
```

## H.2 EXAMPLE/KNOWLEDGE PROMPT FOR TASK DECOMPOSITIONS AND PROCESS CONTROL, AND CONSTRAINT GENERATOR

```
Examples for geometric constraint generation and flow constraint generation for each stage under the task:
  – "pouring liquid from teapot until the cup is filled":
    – "grasp teapot" stage: (stage 1)
      – <"grasp", "the handle of the teapot">
      – <"sub-goal constraints", "the heading direction of the gripper approach of the robot", "the plane of
        ↪  the surface of the table", "the heading direction of the gripper approach of the robot is
        ↪  parallel to the plane of the surface of the table">
      – <"sub-goal constraints", "the heading direction of the gripper binormal of the robot", "the heading
        ↪  direction of the handle of the teapot", "the heading direction of the gripper binormal of the
        ↪  robot is perpendicular to the heading direction of the handle of the teapot">
    – "align teapot with cup opening" stage: (stage 2)
      – <"sub-goal constraints", "the center of the teapot spout of the teapot", "the center of the cup
        ↪  opening of cup", "the center of the teapot spout of the teapot is directly above the
        ↪  center of the cup opening of the cup around 20 centimeters">
    – "tilt teapot until the cup is filled with water" stage: (stage 3)
      – <"sub-goal constraints", "the area of the handle of the teapot", "the normal of the handle of the
        ↪  teapot", "the area of the handle of the teapot rotates around the normal of the handle of the
        ↪  teapot by 30 degress">
      – <"flow constraints", "the cup is filled with water"> (go to stage 3 if satisfied; goto stage 4 if
        ↪  not satisfied)
    – "place the teapot on the table near the cup" stage: (stage 4)
      – <"sub-goal constraints", "the surface of the table", "the center of the cup opening of the cup", "
        ↪  the center of the teapot spout of the teapot is above the surface of the table by 10cm">
      – <"sub-goal constraints", "the center of the body of the teapot", "the center of the body of the cup
        ↪  ", "the distance between the center of the body of the teapot and the center of the body of
        ↪  the cup is around 20cm">
      – <"sub-goal constraints", "the surface of the table", "the plane of the cup opening of the cup", "the
        ↪  surface of the table is parallel to the plane of the cup opening of the cup">

  – "put red block on top of the blue block":
    – "grasp red block" stage:
      – <"grasp", "the body of the red block">
      – <"sub-goal constraints", "the heading direction of the gripper approach of the robot", "the plane of
        ↪  the surface of the table", "the heading direction of the gripper approach of the robot is
        ↪  parallel to the normal of the surface of the table">
    – "drop the red block on top of blue block" stage:
      – <"sub-goal constraints", "the center of the red block", "the center of the blue block", "the center
        ↪  of the red block is directly above the center of the blue block around 20 centimeters">
    – "release the red block" stage:
      – <"release">
  – "open the door around the door hinge":
    – "grasp the door handle" stage:
      – <"grasp", "the handle of the door">
      – <"sub-goal constraints", "the heading direction of the gripper approach of the robot", "the plane of
        ↪  the door of the fridge", "the heading direction of the gripper approach of the robot is
        ↪  parallel to the normal of the door of the fridge">
      – <"sub-goal constraints", "the heading direction of the gripper binormal of the robot", "the heading
        ↪  direction of the handle of the fridge", "the heading direction of the gripper binormal of the
        ↪  robot is perpendicular to the heading direction of the handle of the fridge">
    – "rotate the door" stage:
      – <"sub-goal constraints", "the plane of the surface of the door", "the axis of the hinge of the door
        ↪  ", "the plane of the surface of the door rotates around the axis of the hinge of the door by
        ↪  90 degree">
      – <"path constraints", "the center of the handle of the door", "the axis of the hinge of the door", "
        ↪  the distance between the center of the handle of the robot and the hinge of the body of the
        ↪  door remains unchanged">
    – "release the door" stage:
      – <"release">
  – "cut the cucumber with the kitchen knife":
    – 'grasp the kitchen knife' stage:
      – <"grasp", "the handle of the kitchen knife">
      – <"sub-goal constraints", "the heading direction of the gripper approach of the robot", "the plane of
        ↪  the surface of the table", "the heading direction of the gripper approach of the robot is
        ↪  parallel to the normal of the surface of the table">
      – <"sub-goal constraints", "the heading direction of the gripper binormal of the robot", "the heading
        ↪  direction of the handle of the kitchen knife", "the heading direction of the gripper binormal
        ↪  of the robot is perpendicular to the heading direction of the handle of the kitchen knife">
    – "hang the knife above the cucumber"
      – <"sub-goal constaints", "the center of the blade of the kitchen knife", "the center of the body of
        ↪  the cucumber", "the center of the blade of the kitchen knife is directly above the center of
        ↪  the body of the cucumber by 20 cm">
      – <"sub-goal constaints", "the axis of the cucumber", "the plane of the blade of the knife", "the axis
        ↪  of the cucumber is perpendicular to the plane of the blade of the knife">
      – <"sub-goal constaints", "the heading direction of the blade of the knife", "the plane of the surface
        ↪  of the table", "the heading direction of the blade of the knife is parallel to the plane of
        ↪  the surface of the table">
    – "chop the cucumber" stage:
      – <"path constaints", "the axis of the cucumer", "the plane of the blade of the knife", "the axis of
        ↪  the cucumber remains perpendicular to the plane of the blade of the knife"> (remain from the
        ↪  previous constraints)
      – <"path constaints", "the heading direction of the blade of the knife", "the plane of the surface of
        ↪  the table", "the heading direction of the blade of the knife remains parallel to the plane of
        ↪  the surface of the table"> (remain from the previous constraints)
```

```
1080
1081        – <"sub-goal constaints", "the center of the blade of the kitchen knife", "the center of the surface
1082            ↪ of the table", "the area of the blade of the kitchen knife is above the area of the surface
                ↪ of the table by 1 cm">
           – "release the cucumber" stage:
1083          – <"release">
         – "open the drawer":
1084        – "grasp the drawer handle" stage:
             – <"grasp", "the handle of the drawer">
1085         – <"sub-goal constaints", "the heading direction of the gripper of the robot", "the plane of the
1086            ↪ front face of the drawer", "the heading direction of the gripper of the robot is parallel to
                ↪ the normal of the front door of the drawer">
1087         – <"sub-goal constaints", "the heading direction of the gripper binormal of the robot", "the heading
1088            ↪ direction of the handle of the drawer", "the heading direction of the gripper binormal of the
                ↪ robot is perpendicular to the heading direction of the handle of the drawer">
1089        – "pull the drawer" stage:
             – <"sub-goal constaints", "the center of the handle of the drawer", "the center of the body the
1090            ↪ drawer", "the center of the handle of the drawer move backwards the center of the body of the
                ↪ drawer by around 30 cm">
1091        – "release the drawer" stage:
             – <"release">
1092     – "press the button"
           – "close the gripper" stage:
1093          – <"grasp", "">
           – "move to ready-to-press position" stage:
1094         – <"sub-goal constaints", "the heading direction of the robot approach of the robot", "the center of
1095            ↪ the body of the button", "the heading direction of the gripper approach of the robot colinear
                ↪ with the center of the body of the button">
1096         – <"path constaints", "the heading direction of the gripper of the robot", "the plane of the surface
1097            ↪ of the button", "the heading direction of the gripper of the robot remains parallel to the
                ↪ normal of the surface of the button">
1098        – "pressing" stage:
             – <"sub-goal constaints", "the center of the gripper of the robot", "the center of body of the button
1099            ↪ ", "the center of the gripper of the robot reaches the center of the body of the button">
1100         – <"path constaints", "the heading direction of the gripper of the robot", "the plane of the surface
1101            ↪ of the button", "the heading direction of the gripper of the robot remains parallel to the
                ↪ normal of the surface of the button">
       Example for geometric constraint generation and flow constraint generation under a single stage:
1102   – Orbiting: We can only orbit each time 30 degrees due to design limitation. If we want to orbit for a
1103       ↪ circle, we need to repeatedly orbit 30 degrees by 12 times.
         – "orbit in one circle by x cm"
1104        – <"sub-goal constaints", "the center of A", "the center B", "the center of A orbit the center of B by
               ↪ 30 degrees">
1105        – <"path constaints", "the center of A", "the center of B", "the distance between the center of A and
1106           ↪ the center of B remains x cm">
           – <"flow constaints", "repeat this stage for 12 times (360 degrees in total)">
1107   Flow constraint can be composed together to create complex flow constraint. Like loop-in-a-loop. Since there
1108       ↪ can be ONLY one flow constraint each stage, we need to have standalone stages to place the flow
           ↪ contraint like this:
       Example:
1109   – "..." (stage 3)
1110     – <"flow constaints", "condition"> (go to stage 3 if satisfied; goto stage 4 if not satisfied)
       – "..." (stage 4)
1111     – <"flow constaints", "condition"> (go to stage 3 if satisfied; goto stage 5 if not satisfied)
1112
1113
```

## H.3   SCHEME PROMPT FOR COST FUNCTION GENERATION

```
1114
1115       Please translate all the above geometric constraints and flow constaints for each stage into the Python
              ↪ cost function.
1116   – We can obtain the point cloud by calling Python function "get_point_cloud('the object part' of 'the object
           ↪ '', 'timestamp')".
1117     – we record the position of the 'geometry' since the grasping / contact stage, and record it into array.
1118     – specify 'timestamp' to retrive 'geometry' mask at the given timestamp. For example, timestamp = -2 to
               ↪ retrieve the previous mask at the time of grasping. timestamp = -1 to retrieve the current mask
1119           ↪ .
1120     – Example 1, if I want point cloud of "the axis of the body of the windmill" at its current timestamp, I
               ↪ can obtain the point cloud by "mask = get_point_cloud('the body of the windmill', -1)".
1121     – Example 2, if I want point cloud of "the plane of the surface of the door" at its previous timestamp,
               ↪ I can obtain the point cloud by "mask = get_point_cloud('the surface of the door', -2)".
1122   – Please implement a Python cost function "stage_i_subgoal_constraints()", "stage_i_path_constraints()" for
           ↪ all the constraints tuples in the <> brackets one by one, except for the grasping and releasing
1123       ↪ constraints. It returns the cost measuring to what extent the constraint is satisfied. The
           ↪ constraint is satisfied when the cost goes down to 0.
1124   – Grasping, releasing should be a seperate sub-goal stage.
1125   – Implement "stage_i_flow_constraints()" for the flow constraint if needed, it returns the stage index to
           ↪ transit. If the flow constraints are not specified, we enter the next stage after this stage
1126       ↪ sequentially. Don't call undefined function in the flow constraint !
1127   – For sub-goal constraint 1 of grasping, directly return grasp('the object part' of 'the object').
       – You can specify multiple sub-goal constraints for grasping to specify the approach and binormal.
1128   – For releasing in the sub-goal function directly return release().
       – Constraint codes of each stage are splitted by a line "### <stage constraints splitter> ###"
1129   – The unit of length is meter.
       – The stage start from 1.
1130   – Don't omit stages for the repeating stages, expand and list them one by one.
       – Don't call function of other stage and is not defined, copy the function if necessary, but don't just call
1131       ↪ it.
1132   – Left is -x axis, right is x axis, up is z axis, down is -z axis, front is y axis, back is -x axis.

1133   Here are some examples:
       ### <stage constraints splitter> ### (if any)
       ### stage ? sub-goal constraints
```

```
1134
1135   def stage_?_subgoal_constraint1():
1136     """constraints: <"grasp", "the body of the banana"> """
       return grasp("the body of the banana")

1137
1138   ### <stage constraints splitter> ###
1139   ### stage ? sub-goal constraints
       def stage_?_subgoal_constraint1():
1140     """constraints: <"sub-goal constraints", "the axis of the body of the cucumber", "the plane of the blade
             ↪ of the kitchen knife", "the axis of the body of the cucumber is perpendicular to the plane of
             ↪ the blade of the kitchen knife"> (for cutting cucumber)"""
1141     pc1 = get_point_cloud("the body of the cucumber", -1)
1142     pc2 = get_point_cloud("the blade of the kitchen knife", -1)

1143     # Calculate the axis of the the body of the cucumber (pc1)
       # Compute the covariance matrix of the points in the point cloud
1144     covariance_matrix_cucumber = np.cov(pc1.T)
       # Get the eigenvalues and eigenvectors of the covariance matrix
1145     eigenvalues_cucumber, eigenvectors_cucumber = np.linalg.eig(covariance_matrix_cucumber)
       # The eigenvector corresponding to the largest eigenvalue is the axis of the body of the cucumber
1146     cucumber_axis = eigenvectors_cucumber[:, np.argmax(eigenvalues_cucumber)]
1147     if cucumber_axis[np.argmax(np.abs(cucumber_axis))] < 0:
         cucumber_axis = -cucumber_axis

1148
1149     # Calculate the normal vector of the plane of the blade of the kitchen knife (pc2)
       covariance_matrix_knife = np.cov(pc2.T)
1150     eigenvalues_knife, eigenvectors_knife = np.linalg.eig(covariance_matrix_knife)
       # The eigenvector corresponding to the smallest eigenvalue is the normal vector of the surface
1151     knife_surface_normal = eigenvectors_knife[:, np.argmin(eigenvalues_knife)]
       if knife_surface_normal[np.argmax(np.abs(knife_surface_normal))] < 0:
1152       knife_surface_normal = -knife_surface_normal

1153
1154     # Normalize both vectors
       cucumber_axis = cucumber_axis / np.linalg.norm(cucumber_axis)
       knife_surface_normal = knife_surface_normal / np.linalg.norm(knife_surface_normal)

1155
1156     # Compute the dot product between the cucumber axis and knife surface normal
       dot_product = np.dot(cucumber_axis, knife_surface_normal)

1157
1158     # cucumber_axis perpendicular to knife surface is to be parallel to the knife surface normal
       cost = (1 - abs(dot_product)) * 5.

1159     return cost

1160   def stage_?_subgoal_constraint2():
1161     """constraints: <"sub-goal constraints", "the center of the body of the cucumber", "the center of the
             ↪ body of the kitchen knife", "the center of the body of the cucumber is directly above the
1162          ↪ center of the body of the kitchen knife by 10cm"> (for cutting cucumber)"""
1163     pc1 = get_point_cloud("the body of the cucumber", -1)
       pc2 = get_point_cloud("the body of the kitchen knife", -1)

1164     # Compute the mean position of the body the cucumber and the body of the kitchen knife
1165     body_of_cucumber_center = np.mean(pc1, axis=0)
       body_of_knife_center = np.mean(pc2, axis=0)

1166
1167     # Calculate the horizontal distance (x, y coordinates) between the centers
       horizontal_distance = np.linalg.norm(body_of_cucumber_center[:2] - body_of_knife_center[:2])

1168
1169     # Calculate the center of the body of the knife center should be 20 cm above the center of the body of
             ↪ the cucumber
       vertical_distance = body_of_knife_center[2] - 0.1 - body_of_cucumber_center[2]

1170
1171     cost = abs(vertical_distance) + horizontal_distance

1172     return cost

1173   def stage_?_subgoal_constraint3():
1174     """constraints: <"sub-goal constraints", "the heading direction of the blade of the knife", "the plane
             ↪ of the surface of the table", "the heading direction of the blade of the knife is parallel to
1175          ↪ the plane of the surface of the table"> (for cutting cucumber)"""
1176     pc1 = get_point_cloud("the blade of the knife", -1)
       pc2 = get_point_cloud("the surface of the table", -1)

1177     # Calculate the heading direction vector of the plane of the blade of the knife (pc1)
1178     covariance_matrix_knife = np.cov(pc2.T)
       eigenvalues_knife, eigenvectors_knife = np.linalg.eig(covariance_matrix_knife)
1179     # The eigenvector corresponding to the smallest eigenvalue is the normal vector of the surface
       knife_surface_heading = eigenvectors_knife[:, np.argmin(eigenvalues_knife)]
1180     if knife_surface_heading[np.argmax(np.abs(knife
       _surface_heading))] < 0:
1181       knife_surface_heading = -knife_surface_heading

1182
1183     # Calculate the normal vector of the plane of the surface of the table (pc2)
       covariance_matrix_table = np.cov(pc2.T)
1184     eigenvalues_table, eigenvectors_table = np.linalg.eig(covariance_matrix_table)
       # The eigenvector corresponding to the smallest eigenvalue is the normal vector of the surface
1185     table_surface_normal = eigenvectors_table[:, np.argmin(eigenvalues_table)]
       if table_surface_normal[np.argmax(np.abs(table_surface_normal))] < 0:
1186       table_surface_normal = -table_surface_normal

1187     # Normalize both vectors
       table_surface_normal = table_surface_normal / np.linalg.norm(table_surface_normal)
```

```
1188        knife_surface_heading = knife_surface_heading / np.linalg.norm(knife_surface_heading)

1189
            # Compute the dot product between the table axis and knife surface normal
1190        dot_product = np.dot(table_surface_normal, knife_surface_heading)

1191
            # knife surface heading parallel to the plane of the table surface is to be perpendicular to the table
1192            ↪ surface plane normal
            cost = abs(dot_product) * 5.
1193        return cost

1194    def stage_?_subgoal_constraint1():
            """constraints: <"sub-goal constraints", "the plane of the surface of the door", "the axis of the hinge
1195            ↪ of the door", "the plane of the surface of the door rotate around the axis of the hinge of the
1196            ↪ door by 60 degrees"> (for opening the door)"""
            pc1 = get_point_cloud("the surface of the door", -1)
1197        pc1_previous = get_point_cloud("the surface of the door", -2)
            pc2 = get_point_cloud("the hinge of the door", -2)
1198
            # Step 1: Normalize the axis of the hinge of the door (from pc2)
1199        covariance_matrix_door = np.cov(pc2.T)
            eigenvalues_door, eigenvectors_door = np.linalg.eig(covariance_matrix_door)
1200        door_axis = eigenvectors_door[:, np.argmax(eigenvalues_door)]
            door_axis = door_axis / np.linalg.norm(door_axis)  # Normalize the axis vector
1201        if door_axis[np.argmax(np.abs(door_axis))] < 0:
              door_axis= -door_axis
1202
            # Step 2: Convert the angle from degrees to radians
1203        angle_radians = np.radians(angle_degrees)

1204
            # Step 3: Compute the rotation matrix using Rodrigues' rotation formula
1205        K = np.array([[0, -door_axis[2], door_axis[1]],
                          [door_axis[2], 0, -door_axis[0]],
1206                      [-door_axis[1], door_axis[0], 0]])  # Skew-symmetric matrix for door_axis
            I = np.eye(3)  # Identity matrix
1207        rotation_matrix = I + np.sin(angle_radians) * K + (1 - np.cos(angle_radians)) * np.dot(K, K)

1208
            # Step 4: Rotate each point in pc1
1209        rotated_pc1 = np.dot(pc1_previous - pc2.mean(0), rotation_matrix.T) + pc2.mean(0)  # Apply rotation
1210            ↪ matrix to each point

1211        # Step 5: compute the cost of how pc1 aligns with rotated_pc1.
            cost = np.linalg.norm(pc1 - rotated_pc1, axis=1).sum()
1212        return cost

1213
1214    ### <stage constraints splitter> ###
1215
        ### stage ? sub-goal constraints
1216    def stage_?_subgoal_constraint1():
            """constraints: <"release"> """
1217        release()
            return
1218
        ## Some geometry-related knowledge here:
1219    {}
        ## End knowledge
1220
        Please write the codes below:
1221    ### <stage constraints splitter> ###
        ### stage 1 sub-goal constraints (if any)
1222    ## if it is a grasping constaints
        def stage_1_subgoal_constraint1():
1223        """constraints: <"grasp", "`geometry` of `the object part` of `the object`"> """
            return grasp("`the object part` of `the object `")
1224
1225    def stage_1_subgoal_constraint1():
            """constraints: <?, ?, ?,..., ?>"""
1226        mask1 = get_point_cloud(?)
            mask2 = get_point_cloud(?)
1227        ...
            return cost
1228    # Add more sub-goal constraints if needed
        ...
1229
        ### stage 1 path constraints (if any)
1230    def stage_1_path_constraint1():
            """constraints: <?, ?, ?, ?>"""
1231        mask1 = get_point_cloud(?)
            mask2 = get_point_cloud(?)
1232        ...
            return cost
1233
        # Add more path constraints if needed
1234    ...

        Finally, write a list of "`geometry` of `the object part` of `the object`" in all the <> brackets:
1235    object_to_segment = [?]
```

## H.4 Example/Knowledge Prompt for Cost Function Generation

```
Here are some geometry-related and control-flow-related knowledge:
THE EXAMPLES ARE ONLY FOR YOUR REFERENCE. YOU NEED TO ADAPT TO THE CODE FLEXIBLY AND CREATIVELY ACCORDING TO
    ↪    DIFFERENT SCENARIOS !

# Chapter 1: normal, axis, heading direction, binormal:
- Notice: The largest axis component of the normal / axis / heading direction should always be positive !
- To find the heading direction is the same of finding the axis
- Example:
    """
    Finds the normal (normal vector) of a plate given its point cloud.

    Args:
        pc: numpy array of shape (N, 3), point cloud of the plate.

    Returns:
        plate_normal: A normalized vector representing the normal vector of the plate.
    """
    # Compute the covariance matrix of the point cloud
    covariance_matrix = np.cov(pc.T)

    # Perform eigen decomposition to get eigenvalues and eigenvectors
    eigenvalues, eigenvectors = np.linalg.eig(covariance_matrix)

    # The eigenvector corresponding to the smallest eigenvalue is the normal vector to the plate's surface
    plate_normal = eigenvectors[:, np.argmin(eigenvalues)]
    if plate_normal[np.argmax(np.abs(plate_normal))] < 0:
        plate_normal = -plate_normal

    # Normalize the normal vector
    plate_normal = plate_normal / np.linalg.norm(plate_normal, axis=-1)

    return plate_normal

- Next example:
    """
    Finds the axis of a cylinder given its point cloud.

    Args:
        pc: numpy array of shape (N, 3), point cloud of the cylinder.

    Returns:
        cylinder_axis: A normalized vector representing the axis of the cylinder.
    """
    # Compute the covariance matrix of the point cloud
    covariance_matrix = np.cov(pc.T)

    # Perform eigen decomposition to get eigenvalues and eigenvectors
    eigenvalues, eigenvectors = np.linalg.eig(covariance_matrix)

    # The eigenvector corresponding to the largest eigenvalue represents the axis of the cylinder
    cylinder_axis = eigenvectors[:, np.argmax(eigenvalues)]
    if cylinder_axis[np.argmax(np.abs(cylinder_axis))] < 0:
        cylinder_axis = -cylinder_axis

    # Normalize the axis vector
    cylinder_axis = cylinder_axis / np.linalg.norm(cylinder_axis, axis=-1)

    return cylinder_axis
- To find out the heading direction of long-shaped object, find the max PCA component.
- To find out the normal of a surface, find the min PCA component.
- To find out the axis of an object, there are two cases.
    - For long-shaped object like bolt, carrot, etc., its the max PCA component
    - For fat-shaped object like bowl, nut, etc., its the min PCA component

- A axis / heading direction / normal that is perpendicular to a plane / surface is parallel to the normal.
- A binormal is the vector that is both perpendicular to the axis / heading direction and the normal
- parallel: cost = (1 - np.abs(dot_product)) * 5

# Chapter 2: relative position between two points
- Example 1:
    """
    Measures the cost that point 2 is directly below point 1.

    Args:
        pc1: numpy array of shape (N, 3), point cloud of point 1.
        pc2: numpy array of shape (M, 3), point cloud of point 2.

    Returns:
        cost: a non-negative float representing the extent to which point 2 is directly below point 1.
            The lower the cost, the more point 2 is directly below point 1.
    """
    # Compute the center of mass (mean position) for point 1 and point 2
    point1_center = np.mean(pc1, axis=0)
    point2_center = np.mean(pc2, axis=0)

    # Calculate the horizontal distance (x, y coordinates) between the centers
    horizontal_distance = np.linalg.norm(point1_center[:2] - point2_center[:2])

    # Calculate the vertical distance (z coordinate) between the centers
    vertical_distance = point1_center[2] - point2_center[2]
```

```
1296
1297        # If point 2 is not below point 1, add a large penalty to the cost
           if vertical_distance < 0:
1298           cost = abs(vertical_distance) + horizontal_distance + 1000  # Large penalty for incorrect vertical
1299              ↪ position
           else:
1300           cost = horizontal_distance
1301
           return cost
1302
     - Next example:
1303        """
           Measures the cost that point 2 is directly to the left of point 1 by 10 cm.
1304
           Args:
1305           pc1: numpy array of shape (N, 3), point cloud of point 1.
1306           pc2: numpy array of shape (M, 3), point cloud of point 2.

1307        Returns:
              cost: a non-negative float representing the extent to which point 2 is directly to the left of point
1308             ↪ 1 by 10 cm.
                 The lower the cost, the closer point 2 is to being exactly 10 cm to the left of point 1.
1309        """
           # Compute the center of mass (mean position) for point 1 and point 2
1310        point1_center = np.mean(pc1, axis=0)
1311        point2_center = np.mean(pc2, axis=0)
1312
           # Calculate the horizontal distance (x-axis) between point 1 and point 2
1313        x_distance = point2_center[0] - point1_center[0]
1314
           # Calculate the y and z distances (vertical and depth positions)
           y_distance = abs(point2_center[1] - point1_center[1])
1315        z_distance = abs(point2_center[2] - point1_center[2])
1316
           # The ideal x distance should be -0.10 meters (to the left by 10 cm)
1317        cost = abs(x_distance + 0.10) + y_distance + z_distance  # Sum all deviations from ideal positioning
1318
           return cost
1319
     # Chapter 3: control flow
1320    We use flow constraints for control flow, which specify transitions among different stages.
     - Repetition control flow: Do <something> until some <condition>
1321    - For example:
     <"flow constraint", "Repeat this stage until the box reaches the table edge">
1322    def stage_'i'_flow_constraint1():
       while True:
1323        # query GPT-4O
           query = "Is the box on the table edge? You only need to answer 'yes' or 'no'"
1324        answer = query_GPT(query)
           if answer.strip().lower() == "yes"
1325         return 'i+1' # go to next stage
           else:
1326         return 'i' # repeat this stage to continue pushing the box
     ## Repeat until the cup is being filled, then go to stage 3
1327    ## <"flow constraints", "the cup is filled with water">
1328    def stage_i_flow_constraint1():
       while True:
1329        # query GPT-4O
           query = "Is the water filled in the cup? You only need to answer 'yes' or 'no'"
1330        answer = query_GPT(query)
           if answer.strip().lower() == "yes"
1331         return 'i+1'
           else:
1332         return 'i'
1333
     ## Repeat the stage N times
1334    ## <"flow constraints", "repeat this stage N times">
     def stage_i_flow_constraint1():
1335      # CNT is a global counter variable with default value 0, don't initialize it again!
       if CNT < N:
1336        CNT += 1
         return 'i'
1337      CNT = 0
       return 'i+1'
1338    - You can have multiple flow constraint if necessary. They can create complex flow control. Just think about
                ↪ what you do to write flow control in Python code.
1339    For a example:
     ## <"flow constraints", "repeat this stage N times">
1340    ## <"flow constraints", "condition">
     This is example of loop in a loop. The inner loop repeat the stage N times. The outer loop repeat the inner
1341         ↪ loop until condition is satisfied.
     Another example:
1342    ## <"flow constraints", "repeat this stage N times">
     ## <"flow constraints", "condition">
1343
     # Chapter 4: rotation and orbiting
1344    - To rotate, we use sub-goal constraint to first constraints its rotated position
     ## rotate pc around axis by angle_degrees
1345    def stage_?_subgoal_constraint1():
         pc_previous = get_point_cloud("pc", -2)
1346      pc = get_point_cloud("pc", -1)
         object = get_point_cloud("object", -2) # use -2 to specify the previous object
```

```
covariance_matrix = np.cov(object.T)
eigenvalues, eigenvectors = np.linalg.eig(covariance_matrix)
axis = eigenvectors[:, np.argmax(eigenvalues)]
axis = axis / np.linalg.norm(axis, axis=-1)  # Normalize the axis vector

# Step 3: Convert the angle from degrees to radians
angle_radians = np.radians(angle_degrees)

# Step 4: Compute the rotation matrix using Rodrigues' rotation formula
K = np.array([[0, -axis[2], axis[1]],
              [axis[2], 0, -axis[0]],
              [-axis[1], axis[0], 0]])
I = np.eye(3)  # Identity matrix
rotation_matrix = I + np.sin(angle_radians) * K + (1 - np.cos(angle_radians)) * np.dot(K, K)

# Step 5: Rotate each point in pc1 around object's center
rotated_pc = np.dot(pc_previous - object.mean(0), rotation_matrix.T) + object.mean(0)

cost = np.linalg.norm(rotated_pc - pc, axis=-1).sum()
return cost
```

- To orbit: The orientation of pc is unchanged during orbiting. To calculate the position after orbital
  ↪ translation, we first calculate the position of the center of pc rotating around the axis of the
  ↪ object. Next, we translate the whole pc to the rotated center.

```
def stage_?_subgoal_constraint1():
    pc_previous = get_point_cloud("pc", -2)
    pc = get_point_cloud("pc", -1)
    object = get_point_cloud("object", -2) # use -2 to specify the previous object
    covariance_matrix = np.cov(object.T)
    eigenvalues, eigenvectors = np.linalg.eig(covariance_matrix)
    axis = eigenvectors[:, np.argmax(eigenvalues)]
    axis = axis / np.linalg.norm(axis, axis=-1)  # Normalize the axis vector
    # Step 3: Convert the angle from degrees to radians
    # Step 4: Compute the rotation matrix using Rodrigues' rotation formula
    # Step 5: Rotate each point in pc1 around object's center
    orbital_pc_center = np.dot(pc_previous.mean(0) - object.mean(0), rotation_matrix.T) + object.mean(0)
    orbital_pc = orbital_pc - pc_previous.mean(0) + orbital_pc_center
    cost = np.linalg.norm(rotated_pc - pc, axis=-1).sum()
    return cost
```

- For both rotation and orbiting, if the distance is not specified, we need a path constraint to specify the
  ↪ distance between pc center and the object center remain unchanged (same as the distance of
  ↪ pc_previous center and the object center)

```
def stage_?_path_constraint1():
    pc_previous = get_point_cloud("pc", -2)
    pc = get_point_cloud("pc", -1)
    object = get_point_cloud("object", -2) # use -2 to specify the previous object
    distance_previous = np.linalg.norm(pc_previous.mean(0) - object.mean(0))
    distance = np.linalg.norm(pc.mean(0) - object.mean(0))
    cost = abs(distance_previous - distance)
    return cost
```

- If certain distance 'x' is specified, we need path constraint to remain the specified distance:

```
def stage_?_path_constraint1():
    # get pc, and object
    distance = np.linalg.norm(pc.mean(0) - object.mean(0))
    cost = abs(distance - x)
    return cost
```

- To turn something, rotate all its points around its axis by some angle.
- To orbit in circle, using flow control to repeat this stage 12 times: <"flow constraints", "repeat this
  ↪ stage 12 times">. For sub-goal constraint, orbit by 30 angle_degrees.
- To rotate / orbit clockwisely, the angle is negative; Otherwise, the angle is positive.

```
# Chapter 4: Relationship between points and vector
```
- Colinear: point B colinear with object A's axis / normal / heading direction by distance x if:
  point B = point A's center + normalize(point A's axis / normal / heading direction) * x
- move towards / backwards / against / away:
  - We need to calculate the target point first and calculate the distance between previous point and the
    ↪ target point as the cost
  - points A move towards / to points B by distance:
    previous point A = get_point_cloud(A, -2)
    current point A = get_point_cloud(A, -1)
    moving direction = normalized(vector of previous point A to B)
    target position of point A = points A + moving direction * distance
    cost = np.linalg.norm(target position of point A - current position of point A) ## the cost is
    ↪ calculated based on the distance between target point and current point !!
  - points A move backward / against / away from points B by distance:
    previous point A = get_point_cloud(A, -2)
    current point A = get_point_cloud(A, -1)
    moving direction = normalized(vector of previous point A to B)
    target position of point A = points A + moving direction * distance
    cost = np.linalg.norm(target position of point A - current position of point A) ## the cost is
    ↪ calculated based on the distance between target point and current point

## H.5 SCHEME PROMPT FOR GEOMETRY PARSER

There are totally {number of pair} pair of images.
For each pair, the left image is the image of {object name} with different part highlighted in red. The
↪ right image is the segmentation mask highlighted in white to represent different parts of {object
↪ name}. These images are named as image i, ... (i=0, 1, 2, ...)

Please infer what is highlighted in red for the left image one by one, and then select one of the image for
↪ {geometric part name}.
– Output: image {image_index}, 'geometry' (i=0,1,2... is the index number) at the end in a single line.
– Where 'geometry' is the geometry of object, like the edge, the center, the area, left point, right, point,
↪ etc..

Write a Python function to find out the {geometric part name} given the segmentation of image {object name},
↪ {image_index}.
– the input 'mask' is a boolean numpy array of a segmentation mask in shapes (H, W)
– return the mask which is a numpy array.
– You can 'import numpy as np' and 'import cv2', but don't import other packages
– mask_output should still be in the shape(H, W)
## code start here
def segment_object(mask):
    ...
    return mask_output
Please directly output the code without explanations. Complete the comment in the code. Remove import lines
↪ since they will be manually imported later.

## H.6 KNOWLEDGE PROMPT FOR GEOMETRY PARSER

– To find hinge / axis, output the image of its door, and see which side to segment. For a rotating
↪ object part, the hinge / axis and the handle are of the opposite position. For example, for
↪ finding the hinge of the microwave, output the image of microwave door first. And if the handle
↪ is on the left of the the door, the hinge should locate at the right edge of its door.
– For a sliding body, the slider should be parallel to the edge of the frame.
– sample code to find the complete edge. You need to adjust the code to choose the left / right / top /
↪ bottom edge accordingly. For example, to fine the left edge, find the leftmost True value by
↪ iterating over each row to find the leftmost True value

```python
def find_edges(mask):
    """
    Find the edges of a binary mask using Canny edge detection.

    Parameters:
        mask (np.ndarray): Binary image (mask) with 1s representing the object and 0s representing the
            ↪ background.

    Returns:
        np.ndarray: Edge mask with 255 at the edges of the object and 0s elsewhere.
    """
    # Convert mask to uint8 if not already
    mask = (mask * 255).astype(np.uint8) if mask.max() == 1 else mask

    # Apply Canny edge detection
    edges = cv2.Canny(mask, 100, 200)

    # shift the edge down a little bit !
    edges = np.roll(edges, 3, axis=0)

    # Set the top rows to zero to prevent wrap-around artifacts
    edges[:3, :] = 0

    return edges
```

– return the mask directly if the mask does not need to be processed