# OpenReview forum: "Geometric Constraints as General Interfaces for Robot Manipulation"
_ICLR.cc/2026/Conference — ICLR 2026 Conference Withdrawn Submission_

### Official Review · Reviewer_wsXV · 2025-10-31

**Soundness:** 3
**Presentation:** 3
**Contribution:** 3
**Rating:** 6
**Confidence:** 4

**Summary:**

This paper proposes using VLM-specified geometric constraints as an intermediate representation for robot manipulation tasks. The pipeline consists of the following steps: inferring relevant object geometries using a combination of foundation models (object detectors, segmenters, VLMs), using VLMs to generate constraints as Python funtions that maps the set of identified geometries to a cost for the given task, and an optimization-based motion planner that generates motion trajectories of the robots by solving the constraints. The work builds on top of previous work ReKep, which considers only keypoint-based constraints. The main contribution appears to be the support for a broader set of geometries, including those that are difficult to be specified with keypoints, such as the rim of a cup.

**Strengths:**

- The paper is written with clarity in general and easy to follow.
- The empirical results are extensive across two simulated domains and in the real world and are shown to be competitive to a range of baselines which require task-specific training data.
- The paper also demonstrates a set of additional benefits, including learning from human demonstrations, and adaptation.

**Weaknesses:**

- The provided URL points to an empty google sites. There is no video results.
- While the proposed “select-process” module supports a broader set of geometries compared to keypoints used in ReKep, an apparent issue appears to be that it’s difficult to be integrated in the downstream optimization process. For example, in Figure 4, when a cup rim is identified, how may the VLM write the constraint function to leverage such geometry? This appears to be unclear from the paper.

**Questions:**

See weaknesses section above.

---

### Official Review · Reviewer_KGDf · 2025-11-01

**Soundness:** 2
**Presentation:** 2
**Contribution:** 1
**Rating:** 2
**Confidence:** 5

**Summary:**

This paper proposes the GeoManip framework, which leverages essential geometric constraints derived from object-part relations for robot manipulation. This framework captures geometric constraints and translates them into low-level actions. This paper first proposes a constraint generator to predict stage-specific geometric constraints and a geometry parser to locate the involved object parts. Further, a solver is utilized to optimize trajectories. The results prove the effectiveness of this method.

**Strengths:**

1.  This paper proposes a VLM-driven method for fine-grained robotic manipulation with geometric constraints.
2. This paper proposes a cost function for the trajectory generation.

**Weaknesses:**

1. This paper presents limited novelty compared to ReKep. The method for deriving geometric information still relies on the inherent capabilities of existing models such as SAM and VLMs, and this level of incremental work is insufficient to meet the ICLR acceptance criteria.

2. This method relies on a carefully human-designed prompt and motion planning rule for target task manipulation, which makes it difficult to apply in fine-grained robotic manipulation tasks such as folding.

3. Relying on the static vision perception and motion planning, it seems that this method cannot correct its behavior in a dynamic environment when failure cases occur. It needs to re-prompt with human intervention, re-generate the motion trajectory, which makes it not suitable for dynamic robotic manipulation tasks.

**Questions:**

1. This work utilizes SAM and Grounding-DINO for object detection and segmentation to extract geometric-aware information; however, these perception model always fails when the objects are not obvious, so how does this kind of framework improve the robustness instead of utilizing a more powerful vision foundation model?

---

### Official Review · Reviewer_G14J · 2025-11-01

**Soundness:** 3
**Presentation:** 2
**Contribution:** 3
**Rating:** 4
**Confidence:** 3

**Summary:**

GeoManip tackles the problem of generalizable robotic manipulation by representing tasks through object–part geometric constraints. The authors note two main challenges in prior work: weak transfer of manipulation skills across object categories and the absence of interpretable interfaces linking language to low-level control. To address this, GeoManip introduces a training-free framework consisting of three modules. A Geometry Parser extracts functional object parts via SAM and vision–language reasoning, a Constraint Generator converts task descriptions into symbolic geometric relations, and a Solver optimizes these constraints into executable trajectories. Experiments in Meta-World, OmniGibson, and real-world settings show that GeoManip surpasses existing baselines and supports interactive task editing, failure correction, and demonstration-based learning, demonstrating strong generalization to new objects and tasks.

**Strengths:**

**Originality**

GeoManip introduces the idea of using object–part geometric constraints as a general interface for robotic manipulation, differing from prior approaches that map vision or language directly to actions. Its main contribution lies in integrating part parsing, constraint generation, and trajectory optimization into a unified, training-free framework. However, the paper does not discuss or compare with closely related methods such as CoPa, which share a similar underlying concept, leaving some ambiguity in how its originality stands relative to existing work.

**Quality**

The technical quality of the work is strong, with a well-structured methodology and thorough implementation across both simulation and real-world experiments. The modular design is well justified, and the results clearly demonstrate the feasibility and generalization ability of the approach. Although some quantitative evaluations are relatively limited, the experiments as a whole adequately support the paper’s main claims.

**Clarity**

The paper is well organized and easy to follow, with clear figures and explanations that effectively convey the system workflow and constraint mechanism. The writing is concise, and the main technical ideas are communicated clearly. Some implementation details such as prompt construction and parameter settings are described briefly, but this does not significantly affect the overall readability.

**Significance**

GeoManip provides an interpretable and general framework for connecting language, geometry, and robot action through spatial constraints. Its training-free design and demonstrated transfer across diverse tasks highlight both practical potential and conceptual impact. The approach contributes meaningfully to advancing geometry-based reasoning in robotic manipulation and may inspire further work on modular, explainable robot learning.

**Weaknesses:**

**Limited Real-Time Efficiency and Computational Cost**

The GeoManip framework relies on multiple sequential vision–language model calls for segmentation, constraint generation, and cost function formulation, followed by trajectory optimization. This design introduces significant computational overhead, yet the paper does not report inference time or system latency. Without runtime analysis, it remains unclear whether the framework can operate effectively in time-sensitive or dynamic environments, where high responsiveness and low-latency control are essential.

**Limited Expressiveness of Geometric Constraints**

GeoManip primarily models static spatial relations between rigid parts, such as position and orientation. However, many real-world tasks involve non-rigid objects like cloth, rope, or deformable materials, as well as dynamic interactions with friction, elasticity, or moving targets. The current constraint formulation lacks the expressiveness to handle these phenomena, limiting the framework’s applicability to more complex or physics-rich manipulation tasks.

**Insufficient Baseline Coverage and Fairness**

The experimental evaluation omits several relevant comparisons that would strengthen the analysis. In OmniGibson, GeoManip is only compared with ReKep, and in Meta-World it is constrained to simple translational motions. The real-world tests include only OpenVLA as a baseline. The absence of comparisons with classical or constraint-based planners such as LGP, CHOMP, or CuRobo makes it difficult to assess relative advantages in constraint feasibility, trajectory optimality, and computational efficiency.

**Lack of Ablation and Model Dependency Analysis**

The method’s performance heavily depends on large vision-language models for reasoning and part parsing, yet the paper does not include ablation experiments to quantify this dependence. No analysis is provided on how results change when using smaller or weaker models (e.g., replacing GPT-4o with a lightweight LLM), making it hard to judge the robustness and scalability of the proposed approach.

**Unclear Relation to Prior Constraint-Based Manipulation Work**

The paper positions “geometric constraints as a general interface” as its key contribution but does not discuss or cite conceptually similar research, such as "CoPa: General Robotic Manipulation through Spatial Constraints of Parts". Both works share the core idea of using part-level spatial constraints to generalize manipulation behavior. The lack of acknowledgment or empirical comparison with CoPa leaves the contribution boundary ambiguous and reduces clarity in positioning within the existing literature.

**Questions:**

**Questions**

**Q1: Generalization and Robustness Boundary** – The paper emphasizes strong out-of-distribution generalization, yet it remains unclear how GeoManip performs when facing geometrically novel objects that are linguistically describable but structurally unseen. Can the vision–language model accurately infer appropriate manipulation constraints for such unfamiliar geometries? In addition, how well does the system maintain performance under environmental variations such as changes in lighting, occlusion, or clutter?

**Q2: Inference Speed and Computational Efficiency** – What is the total runtime required for a full manipulation task in both simulation and real-world experiments? Please provide detailed timing statistics for each pipeline component—part parsing, constraint generation, cost function formulation, and trajectory optimization—to assess whether the framework can meet real-time or on-robot control requirements.

**Suggestions**

**S1: Broader Generalization and Robustness Evaluation** – Evaluate GeoManip on linguistically describable but geometrically novel objects, as well as under diverse environmental variations such as lighting, occlusion, clutter, and background changes. Provide quantitative success rates or degradation curves to clearly define the system’s generalization boundary and real-world robustness.

**S2: Comprehensive Runtime Profiling** – Report detailed timing statistics for each pipeline stage, including part parsing, constraint generation, cost function computation, and trajectory optimization, in both simulation and real-world settings. This would clarify computational feasibility and identify potential bottlenecks for real-time or on-robot deployment.

**S3: Fine-Grained Module Diagnostics** – Introduce explicit quantitative metrics for each system component, such as the usable part detection rate in the select–process stage, constraint correctness or consistency score, and average error of cost function optimization. These metrics would help isolate dominant error sources and guide targeted system improvements.

---

### Note · Authors · 2025-11-13

I have read and agree with the venue's withdrawal policy on behalf of myself and my co-authors.